# Antifungal Activities of Multi-Halogenated Indoles Against Drug-Resistant *Candida* Species

**DOI:** 10.3390/ijms262210836

**Published:** 2025-11-07

**Authors:** Hyeonwoo Jeong, Bharath Reddy Boya, Yong-Guy Kim, Jin-Hyung Lee, Jintae Lee

**Affiliations:** School of Chemical Engineering, Yeungnam University, Gyeongsan 38541, Republic of Korea; wawdgud@yu.ac.kr (H.J.); yongguy7@ynu.ac.kr (Y.-G.K.)

**Keywords:** antifungal, biofilm, *Candida*, halogenated indoles, hyphae

## Abstract

The emergence of drug-resistant *Candida* species has created an urgent need for non-toxic molecules that inhibit fungal growth, biofilm development, and hyphal formation. In this study, fifty multi-halogenated indole derivatives were screened against ten *Candida* species, including azole-resistant *C. albicans*, *C. auris*, *C. glabrata,* and *C. parapsilosis*. Among them, 4,6-dibromoindole and 5-bromo-4-chloroindole exhibited the strongest antifungal and antibiofilm effects, with minimum inhibitory concentration (MIC) values of 10–50 µg/mL, outperforming ketoconazole and comparable to miconazole. Both di-halogenated indoles markedly inhibited cell aggregation, yeast-to-hyphae transition, and induced reactive oxygen species (ROS) accumulation, contributing to fungicidal activity. Microscopic analyses revealed the disruption of hyphal networks and reduced biofilm biomass. They showed moderate cytotoxicity in human hepatocellular carcinoma (HepG2) cells (median lethal dose, LD_50_ = 35.5 µg/mL and 75.3 µg/mL) and low phytotoxicity in plant assays. The quantitative structure–activity relationship (QSAR) model identified halogen substitution at C4, C5, and C6 positions as optimal for antifungal activity due to enhanced hydrophobic and electron-withdrawing effects. Together, these findings demonstrate that di-halogenated indoles serve as potent, low-toxicity inhibitors of *Candida* growth, biofilms, and morphogenesis, providing a promising scaffold for next-generation antifungal agents targeting drug-resistant *Candida* species.

## 1. Introduction

Invasive and mucosal infections caused by *Candida* species remain a major global health concern, especially among immunocompromised patients [1]. Although *C. albicans* predominates, non-albicans species such as *C. auris*, *C. glabrata*, and *C. parapsilosis* are emerging with intrinsic or acquired resistance to azoles and echinocandins [2,3].

Biofilm formation and morphogenetic plasticity are central to *Candida* pathogenicity [4,5,6,7]. Biofilms act as physical and metabolic barriers that reduce drug penetration and enable the formation of “persister-like” cells highly tolerant to antifungal therapy [8]. Moreover, the yeast-to-hyphae transition by the expression of adhesins (*ALS1* and *ALS3*) and hyphal wall proteins (*HWP1* and *ECE1*) enhance colonization and invasion of host tissues [9]. *ALS3* is vital for *C. albicans* clearance in mouse [10]. Because these biofilm-embedded populations exhibit metabolic dormancy, small molecules that suppress biofilm formation or hyphae development are increasingly viewed as promising alternatives to fungicidal drugs [11,12,13].

Halogenation is a powerful approach in medicinal chemistry that enhances lipophilicity, metabolic stability, and molecular recognition through halogen bonding and electronic modulation [14,15,16,17,18]. About one-quarter of approved drugs contain halogen atoms, reflecting their pharmacological advantages. A recent study showed that halogenation strengthens antimicrobial potency and resistance-evasion capacity across multiple scaffolds, including phenols, quinolones, and indoles [14]. In fungi, halogenated phenolics [19], chalcones [20], chromones [21], salicylanilides [22], quinones [23], and pyrrolopyrimidines [24] have shown notable antifungal or antibiofilm activity, often mediated by inhibition of yeast-to-hyphae transition and oxidative stress [25].

Indole is a scaffold widely distributed in natural metabolites as a cross-kingdom signaling modulator and synthetic drugs [26]. Recent studies have highlighted indole as a versatile scaffold in modern drug discovery owing to its broad pharmacological profile, including anticancer, anti-inflammatory, and anti-infective properties [27,28,29,30,31]. Particularly, halogenated indoles have demonstrated broad-spectrum antimicrobial activity [32]. Among dozens of mono-halogenated indoles, 5-iodoindole exhibited the strongest activity, eradicating bacterial persisters and inhibiting biofilm formation of *Escherichia coli* and *Staphylococcus aureus* [33], *Acinetobacter baumannii* [34], and *Serratia marcescens* [35]. Also, 7-benzyloxyindole and methylindoles suppressed *Candida albicans* filamentation and biofilm formation [36,37,38,39]. However, most reports focused on mono-halogenated derivatives, and the effects of multiple halogen substitutions on antifungal efficacy and safety remain poorly understood. Recently, multi-halogenation has been shown to markedly influence molecular electronics and redox potential, improving target binding and ROS-generating ability while maintaining low chemical toxicity [40,41]. These observations suggest that multi-halogenated indoles could further enhance antifungal and antibiofilm activities [42].

In this study, 50 multi-halogenated indole derivatives were investigated against ten important *Candida* species, including drug-resistant strains of *C. albicans*, *C. auris*, *C. glabrata*, and *C. parapsilosis*. Two di-halogenated indoles were identified with potent antifungal and antibiofilm activity, examined their effects on hyphal morphogenesis and oxidative stress, and established structure–activity relationships using QSAR and molecular docking analyses. The results reveal that halogenation at specific indole positions led to strong inhibition of *Candida* biofilm formation and morphogenesis with minimal mammalian or plant toxicity.

## 2. Results

### 2.1. Antifungal and Antibiofilm Activities of Multi-Halogenated Indoles

To evaluate the antifungal potential of multi-halogenated indole derivatives, a library of fifty compounds bearing mono-, di-, and tri-halogen substitutions at various ring positions was screened against azole-resistant *C. albicans* DAY185 strain. The backbone indole (#1) and the previous most active mono-halogenated indole 5-iodoindole (#2) [33] displayed MICs of 750 μg/mL and 75 μg/mL, respectively (Figure 1A). In contrast, various di-halogenated indoles demonstrated markedly enhanced antifungal activity compared to 5-iodoindole. For example, two dozen di-halogenated indoles showed MICs of ≤50 μg/mL. Among them, 4,6-dibromoindole (#18) and 5-bromo-4-chloroindole (#24) were identified as the most potent with MICs of 25 μg/mL.

Their antibiofilm activity was evaluated at 20 and 50 μg/mL against *C. albicans* DAY185 strain (Figure 1B). While backbone indole and 5-iodoindole did not affect the biofilm formation at the tested concentrations, several di-halogenated indoles significantly inhibited biofilm formation. Notably, #5, #12, #17, # 18, #19, #24, #29, #33, #40, and #46 at 20 μg/mL markedly inhibited biofilm formation relative to untreated controls. The ten hit compounds were all di-halogenated indoles, bearing F, Cl, Br, or I substitutions at the C4, C5, and C6 positions of the indole ring, with MIC values ≤ 50 μg/mL (Figure 1A,B). The antibiofilm activity of the di-halogenated indoles appears primarily due to their antifungal activity. Among them, the most active compounds, 4,6-dibromoindole and 5-bromo-4-chloroindole, were selected for further analysis.

More detailed biofilm inhibitory assay was performed using 4,6-dibromoindole and 5-bromo-4-chloroindole, with the antifungal agents, ketoconazole and miconazole included for comparison. As shown in Figure 2A–D, the two di-halogenated indoles dose-dependently inhibited biofilm formation, whereas ketoconazole and miconazole were less effective. Planktonic cell growth was also measured, and the results (Figure 2E,F) were consistent with the MICs data (Figure 1A). Ketoconazole had little effect on cell growth (Figure 2G), whereas miconazole completely inhibited growth at 50 μg/mL (Figure 2H). To determine whether the active halogenated indoles exert fungicidal or fungistatic effects, cell viability was assessed by colony-forming unit (CFU) counting. Treatment with 4,6-dibromoindole and 5-bromo-4-chloroindole at 100 µg/mL resulted in complete killing after 4 h of exposure (Figure 2I,J), showing a killing profile comparable to that of miconazole (Figure 2L). These results suggest that two di-halogenated indoles, particularly those bearing bromine and chlorine at the C4, C5, and C6 adjacent positions, exhibit rapid fungicidal activity.

Antifungal activities of two active indoles were further measured against various *Candida* species (Table 1). Both di-halogenated indoles exhibited MIC values ranging from 10 to 50 µg/mL against all ten *Candida* strains. Their activities were significantly stronger than ketoconazole (MIC 25–400 µg/mL) and comparable to miconazole (MIC 10–50 µg/mL). Overall, two active indoles displayed broad-spectrum anti-*Candida* activity against ten *Candida* species, including those less susceptible to azole antifungals.

### 2.2. Antibiofilm and Anti-Hyphal Activities of Active Di-Halogenated Indoles

A live cell imaging system was used to visualize the biofilm inhibition. Consistent with the quantitative biofilm data (Figure 2A,B), the two active hits inhibited biofilm formation in a dose-dependent manner (Figure 3). Through 2D and 3D imaging, both compounds, at 20 and 50 µg/mL, almost completely inhibited the biofilm formation of *C. albicans*, while 5-bromo-4-chloroindole at 10 µg/mL was more active than 4,6-dibromoindole.

The yeast-to-hyphae transition in *C. albicans* represents a key virulence mechanism enabling tissue invasion and biofilm maturation [43]. Hence, the effects of two active indoles on colony morphology, cell aggregation, and hyphal formation were investigated. 5-Bromo-4-chloroindole at 10 and 20 µg/mL markedly inhibited filamentous colony formation on solid agar plates over 5 days, whereas the effect of 4,6-dibromoindole was less distinctive (Figure 4A). At 50 µg/mL, both compounds completely inhibited growth, resulting in no colony formation. Both active indoles at 20 µg/mL markedly inhibited hyphae formation in liquid medium (Figure 4B,C). This inhibition of the yeast-to-hyphae transition was consistent with their biofilm inhibition activity.

### 2.3. Induction of ROS Production and Combinatorial Assay

To understand the possible antifungal mechanism, the effects of the two hits on intracellular ROS levels were investigated, with miconazole used as a positive control. Both compounds (10–50 µg/mL) dose-dependently increased ROS production in *C. albicans*, whereas miconazole induced a significantly higher level of ROS (Figure 5A–C). This result indicates that ROS induction by the active indoles contributes partially to their fungicidal mechanism.

The combinatorial effects of the two best indoles along with six antifungal agents such as polyenes (amphotericin B), azoles (miconazole and econazole) and echinocandins (anidulafungin, caspofungin, and micafungin) were evaluated against *C. albicans* for their impact on planktonic growth and biofilm formation. Combinations with the polyene and the three echinocandins showed indifferent interactions, whereas combinations of the two indole hits with azoles exhibited additive effects (FICI = 0.8–0.9) (Figure 6A,B). The combinatorial treatments exhibited a similar pattern against biofilm formation (Appendix A). Collectively, the two active indoles showed additive interactions with azoles, allowing for lower effective antifungal doses, particularly against azole-resistant *Candida* species.

### 2.4. Structure–Activity Relationship of Halogenated Indoles

To identify the structural motifs responsible for antifungal activity among the tested halogenated indoles, the atom-based 3D-QSAR model was constructed with MIC values against *C. albicans* (Figure 1A). Among the models evaluated, PLS (partial least squares) factor model 4 was optimal, achieving an R^2^ of 0.6556 and a Q^2^ of 0.6726, indicating good statistical validity and predictive ability (Appendix A). The model exhibited a low RMSE (0.29), and the associated *p*-value (1.04 × 10^−5^) indicated statistical significance. Although the stability value was modest (0.391), the Pearson correlation coefficient (r = 0.8318) showed a strong correlation between observed and predicted pMIC values. The scatter plot confirmed this observation (Figure 7A), showing predicted values closely aligned with the experimental data, thereby supporting the model’s predictive accuracy. Contour visualization indicated that substitution at positions C4, C5, and C6 of the indole was favorable (blue contours) for antifungal activity, whereas substitutions at position C2, C3, and C7 were unfavorable (red contours) (Figure 7B). Hydrophobic and electron-withdrawing groups at positions C4 and C6 significantly enhanced antifungal activity. In particular, bromine and chlorine at these positions were most favorable, consistent with the high antifungal activity observed for 4,6-dibromoindole and 5-bromo-4-chloroindole. In contrast, substitution at position C3 decreased antifungal activity, likely due to indole-2,3-dione (isatin) moiety. Notably, tri-halogenated indoles provided no additional improvement in activity and, in some cases, slightly reduced antifungal efficacy (Figure 1A), possibly due to excessive lipophilicity that limited solubility or cellular uptake.

### 2.5. Toxicity Assay in HepG2 Cells and a Plant Model

To evaluate cytotoxicity, human HepG2 hepatocellular carcinoma cells and *Brassica rapa* seedlings were employed. The MTT assay showed that both 4,6-dibromoindole and 5-bromo-4-chloroindole exhibited moderate cytotoxicity toward HepG2 cells, maintaining high viability even at concentrations well above their antifungal MIC. After 24 h of exposure, both compounds exhibited a dose-dependent reduction in cell viability, with an LD_50_ of 36 ± 3 µg/mL and LD_50_ of 75 ± 4 µg/mL, respectively (Figure 8A,B). Notably, cell viability remained above 90% at 20 µg/mL, which is their MIC values against *C. albicans.*

To evaluate the environmental safety, phytotoxic effects of the compounds were tested on plant seedlings. Seed germination rate was not affected up to 50 µg/mL of compounds (Figure 8C). However, plant growth showed a reduction at higher concentrations (Figure 8D,E). Notably, both compounds at 10 and 20 µg/mL enhanced plant growth. These intriguing results suggest that appropriate use of di-halogenated indoles may have potential applications in agriculture. Together, these findings demonstrate that the two di-halogenated indoles below their MIC values, exhibit low mammalian cytotoxicity and favorable plant safety profiles, supporting their potential for therapeutic use and eco-safe applications.

## 3. Discussion

The rise of fungal infections caused by multidrug-resistant *Candida* species has highlighted the urgent need for new antifungal strategies beyond conventional agents. Current drugs such as polyenes, azoles, and echinocandins are increasingly limited by biofilm-associated tolerance, efflux-pump overexpression, and target-site mutations [44,45]. Their hepatotoxicity and nephrotoxicity further constrain therapeutic use [46]. Hence, identifying small molecules that attenuate *Candida* virulence or biofilm formation with minimal host toxicity represents a promising route for next-generation antifungal development [12]. In this study, it was found that multi-halogenation of the indole core markedly enhanced antifungal potency, with two di-halogenated derivatives exhibiting fungicidal activity. These compounds repressed hyphal morphogenesis, biofilm formation while inducing ROS production.

Halogen atoms with their electronegativity and polarizability affect the pharmacological behavior of organic scaffolds. Fluorine can modulate metabolic stability and acidity, while chlorine, bromine, and iodine enhance hydrophobic interactions and van der Waals contacts with biological targets [14,15]. In current results, adjacent di-brominated and bromine–chlorine dual substitutions showed the strongest antifungal effects (Figure 1A), suggesting that increased polarizability and hydrophobic surface area favor antifungal activity. In contrast, tri-halogenation offered no further improvement, suggesting an optimal electronic balance is required for effective antifungal activity. Antifungal screening (Figure 1) and QSAR analyses (Figure 7) revealed that di-halogenation specifically within the C4–C6 region of the indole ring establishes an electronically optimized pharmacophore that enhances membrane-penetrating lipophilicity and redox reactivity, thereby driving potent fungicidal activity and suppression of biofilm-associated morphogenesis.

This trend parallels observations in other halogenated scaffolds such as halogenated chromones [21], salicylanilides [22], quinones [23], and pyrrolopyrimidines [24]. Consistently, the antimicrobial activity of mono-halogenated indoles against *E. coli* followed the hierarchy I > Br > Cl > F (or unhalogenated), particularly when substitutions occurred at the C4–C6 positions of the indole ring [47,48]. Two active compounds, 4,6-dibromoindole and 5-bromo-4-chloroindole, exhibited potent antifungal activity against multiple *Candida* species, including *C. albicans*, *C. auris*, *C. glabrata,* and *C. parapsilosis*. Their MIC values were 25 µg/mL against *C. albicans* and 10–50 µg/mL against non-albicans species, including *C. auris*, a pathogen notorious for its resistance to most clinical antifungals (Table 1). Previous studies of mono-halogenated indoles reported antimicrobial activities against a range of bacteria, such as *Pseudomonas aeruginosa* [49], *E. coli* and *Staphylococcus aureus* [33], *Serratia marcescens* [35], *Acinetobacter baumannii* [34], and pathogenic *E. coli* [47,48]. Collectively, these findings suggest that multi-halogenated indoles possess strong potential as broad-spectrum antimicrobial and antifungal agents.

The yeast-to-hyphae transition underlies tissue invasion and biofilm architecture in *C. albicans* [43]. Microscopic observations showed that di-halogenated indoles delayed or abolished Spider-agar colony formation and hyphal switching (Figure 3). The halogenated indoles did not synergize with amphotericin B (ergosterol-binding) or echinocandins (β-glucan synthase inhibitors) (Figure 6), indicating their primary action is different from ergosterol biosynthesis and β-glucan cell wall synthesis. These alterations may interfere with the Ras1–cAMP–Efg1 signaling cascade, a master regulator of morphogenesis and virulence in *Candida* [50]. Because ROS can modulate cAMP signaling, the oxidative stress induced by halogenated indoles (Figure 5) may indirectly repress Efg1 activity, thereby maintaining cells in a yeast-locked state. ROS accumulation may also compromise mitochondrial membrane potential, disrupting ATP synthesis and biosynthetic capacity. Other indole analogs also showed a similar pathway of inhibition. Indole propionic acid modulates mitochondrial respiration, leads to mitochondrial depolarization, Ca^2+^ overload, and ROS accumulation in *C. albicans*, culminating in apoptosis [51]. Indole-3-carbinol disrupts membrane integrity or perturb membrane dynamics by increased ROS, ultimately causing cell cycle arrest at the G2/M phase [52]. While mitochondrial respiration is not cited in the study as a direct cause, it is well known that mitochondrial respiration is essential for G2/M phase progression. Another study demonstrated novel indole derivatives to have anti-fungal effect of *C. albicans*, while also showing anti-virulence effects by Ras1–cAMP–PKA inhibition [53]. Although not directly measured in this study, prior works with 7-benzyloxyindole and methylindoles exhibited antifungal and antibiofilm activities [36,37] probably due to ROS production. Together, these findings suggest that controlled redox imbalance is a key determinant of halogenated indole bioactivity.

In HepG2 cells, both active compounds maintained >80% viability at twice their MICs, with LD_50_ values of 35.5 and 75.3 µg/mL, respectively (Figure 8). The attenuated mammalian toxicity of di-halogenated indoles compared with mono-halogenated analogs may be attributed to their lower aqueous solubility and reduced electrophilic reactivity [54,55,56]. Also, the compounds showed negligible phytotoxicity, sustaining >90% *Brassica rapa* germination and normal root growth at concentrations up to 20 µg/mL. Although elemental halogens are inherently toxic and highly reactive, a recent study employing a machine learning–based toxicity prediction model indicated a low risk of hepatotoxicity and cardiotoxicity for several halogenated compounds, including indole derivatives [40]. Future studies will expand cytotoxicity profiling to the full library of halogenated indoles in mammalian and plant models to better define structure–toxicity relationships and guide scaffold optimization.

This study expands that multi-halogenated indoles achieve potent antifungal and antivirulence effects through a redox-signaling mechanism. The indole nucleus provides an important scaffold for fine electronic modulation via halogenation, enabling cross-kingdom activity from bacteria to fungi. As a perspective, the dual antifungal and antivirulence profile of halogenated indoles opens multiple applications as topical antifungal formulations for treating mucocutaneous candidiasis and device-related infections (catheters, dentures, prosthetics), surface coatings for antifouling with biocompatible polymers, adjuvant therapy, and plant protection against fungal pathogens such as *Botrytis* or *Fusarium*. Future work should evaluate pharmacokinetics, in vivo efficacy, and mechanistic targets, such as mitochondrial dehydrogenases or redox-sensitive transcription factors. It is also possible that indole scaffold allows introduction of additional functional groups (hydroxyl, methoxy, sulfonamide) to improve bioactivity, solubility or selectivity.

## 4. Materials and Methods

### 4.1. Chemicals, Microbial Strains and Culture Conditions

A library of fifty halogenated indole derivatives (Appendix A) was obtained from Combi-Blocks (San Diego, CA, USA) (≥95% purity). To prepare stock solutions, compounds were dissolved to 100 mg/mL in dimethyl sulfoxide (DMSO), aliquoted, and kept at −20 °C. In all experiments, DMSO was maintained at ≤0.1% (*v*/*v*), which ensured no impact on *C. albicans* growth or biofilm formation. Amphotericin B, anidulafungin, caspofungin, econazole, fluconazole, ketoconazole, micafungin, and miconazole served as positive antifungal controls. Based on preliminary screening, 4,6-dibromoindole and 5-bromo-4-chloroindole were selected as representative multi-halogenated leads.

Ten *Candida* species were obtained from three repositories: ATCC (American Type Culture Collection), KCCM (Korean Culture Centre for Microorganisms) and KCTC (Korean Collection for Type Cultures). The strain panel comprised *C. albicans* DAY185 (fluconazole resistant; MIC > 1024 µg/mL), *C. albicans* ATCC 10231, *C. auris* (KCTC 17809 and KCTC 17810), *C. glabrata* (ATCC 2001, KCCM 12552 and KCCM 50701), and *C. parapsilosis* (ATCC 7330, ATCC 22019 and KCCM 50030). All strains were streaked on potato dextrose (PDA) plates, and a single colony was inoculated into the appropriate medium: potato dextrose broth (PDB) for *C. albicans*, yeast malt broth supplemented with 2% glucose for *C. glabrata* and *C. parapsilosis*, and tryptic soy broth for *C. auris*. Cultures were incubated at 37 °C and at 250 rpm.

### 4.2. Antifungal Assay

The minimum inhibitory concentrations (MICs) were determined with minor modifications to a previously described protocol [24]. *C. albicans* cultures (~10^6^ cells/mL) were diluted in the appropriate medium and exposed to serial concentrations of the test compounds in 96-well plates. Optical density at 600 nm was read with a Multiskan SkyHigh plate reader (Thermo Fisher Scientific, Waltham, MA, USA) for planktonic cell growth and MIC. The MIC was the lowest concentration preventing visible growth after 24 h of incubation. All assays were performed in triplicate on three separate days.

For the fungistatic or fungicidal characteristics, *C. albicans* DAY185 cultures (~10^6^ cells/mL) were diluted in PDB and exposed to the test compounds (0, 20, and 100 µg/mL). At 0, 1, 4, and 24 h after treatment, serially diluted tenfold in sterile phosphate-buffered saline (PBS), and 100 µL was spread onto PDB plates. The plates were maintained without agitation at 37 °C for 24 h, then colony-forming units (CFU) counts were obtained and summarized as log10 CFU/mL.

### 4.3. Biofilm Assay

Biofilms were formed in 96-well plates as described previously [24]. Briefly, two-day-old cultures of *C. albicans* DAY185 were diluted to 1 × 10^6^ cells/mL in PDB medium, and 300 µL was added per well with and without tested chemicals. After 24 h incubation at 37 °C under static conditions, wells were washed thrice with dH_2_O to eliminate non-adherent cells and staining of the remaining biofilms was performed using 0.1% crystal violet (CV, Sigma-Aldrich, St. Louis, MO, USA) for 20 min and rinsed with dH_2_O thrice, and the bound CV was dissolved using 95% ethanol. The absorbance of the dissolved stain was measured at 570 nm in a microplate reader. All assays were performed using two independent cultures, with each condition tested in triplicate.

### 4.4. Microscopic Observation of Candida Biofilms

To observe the antibiofilm activity of the selected halogenated indoles, biofilms of *C. albicans* DAY185 (~10^6^ cells/mL) were formed in PDB at 37 °C for 24 h as stated above. Post static culture, planktonic cells were removed by three gentle phosphate-buffered saline (PBS, pH 7.4) washes, and adherent biofilms were visualized using the iRiS™ digital imaging platform manufactured by Logos BioSystems (Anyang, Republic of Korea). The captured images were then processed and reconstructed into both two-dimensional and three-dimensional color-coded biofilm structures using ImageJ software (version 1.53t, NIH, Bethesda, MD, USA), as previously reported [24].

The SEM analysis was performed as previously described [24]. In brief, *C. albicans* DAY185 cells were diluted to ~10^6^ CFU/mL and treated with or without selected indoles (0, 5, 20 or 50 μg/mL). 300 μL of the treated *C. albicans* cells were dispensed into 96-well plates containing sterile nylon filter membranes (0.4 × 0.4 cm^2^). Following a 24 h incubation at 37 °C under non-shaking conditions, biofilms attached to membranes were fixed using 2.5% glutaraldehyde and 2% formaldehyde. The fixed samples were then gradually dehydrated using a graded ethanol series and subsequently processed through critical point drying with the HCP-2 system (Hitachi, Tokyo, Japan). After platinum sputter-coating, imaging was performed with an ultra-high-resolution field-emission scanning electron microscope (UHR-FESEM, SU8600; Hitachi, Tokyo, Japan) at accelerating voltages of 15 kV.

### 4.5. Hyphal Morphogenesis Analyses

Colony morphology was investigated as previously reported [36]. In brief, *C. albicans* DAY185 was streaked onto solid PDA plates with or without selected halogenated indoles (0, 5, 20, or 50 µg/mL), incubated at 37 °C under static conditions and the colony was observed using an iRiS™ Digital Cell Imaging System for 7 days.

Cell aggregation and hyphae formation were observed as previously described [24]. Briefly, PDB medium was inoculated at a dilution of ~10^6^ CFU/mL with or without halogenated indoles (0, 5, 20 or 50 µg/mL) and cultures were incubated under static conditions at 37 °C for 24 h. Post incubation, hyphal formation and cell aggregation were observed.

### 4.6. Reactive Oxygen Species (ROS) Assay

*C. albicans* DAY185 cells were cultured to the logarithmic phase (4 h), harvested, and resuspended in PBS to 10^5^ CFU/mL. The cells were then treated with halogenated indoles or miconazole (0, 5, 20, or 50 µg/mL) for 1 h at 37 °C with shaking at 250 rpm. After treatment, 20 µM of 5(6)-carboxy-2′,7′-dichlorofluorescein was added to the cell suspensions, transferred to black microtubes and incubated at 37 °C for 30 min under static conditions. Fluorescence intensity (FI) was measured on a multimode microplate reader with excitation/emission set to 506/524 nm. FI values were normalized by cell density (OD_600_).

### 4.7. Combinatorial Assay with Antifungal Drugs

Combinatorial assay was performed with selected indoles and six antifungal drugs using the checkerboard method [57]. The antifungal drugs, such as amphotericin B (MIC = 1 µg/mL), miconazole (MIC = 50 µg/mL), econazole (MIC = 50 µg/mL), anidulafungin (MIC = 0.01 µg/mL), caspofungin (MIC = 0.05 µg/mL), or micafungin (MIC = 0.3 µg/mL) were first added at various concentrations. The mixtures were then aliquoted into microtubes, and the indoles were added at corresponding concentrations to create drug–compound combinations. Next, 300 µL of each combination per well was plated in a 96-well plate and incubated at 37 °C for 24 h in static conditions. After incubation, planktonic cell growth (MIC) and biofilm formation were assessed as mentioned above. The MIC and MBIC were the lowest concentrations that fully inhibited planktonic cell growth and biofilm formation. The interpretation of the fractional inhibitory concentration index (FICI) was based on previously established criteria.$$\mathrm{FICI}=\frac{{\mathrm{MIC}}_{A\;\mathrm{in}\;\mathrm{combination}}\;}{{\mathrm{MIC}}_{A\;\mathrm{alone}}}+\frac{{\mathrm{MIC}}_{B\;\mathrm{in}\;\mathrm{combination}}\;}{{\mathrm{MIC}}_{B\;\mathrm{alone}}}$$

The FIC index (FICI), calculated as the sum of the FICs of drug A and drug B, was interpreted according to prior reports as synergistic when FICI was 0.5 or less, additive when greater than 0.5 but not exceeding 1.0, indifferent when between 1.0 and 2.0, and antagonistic when equal to or greater than 2.0.

### 4.8. QSAR Modeling

An atom-based 3D-QSAR model was developed for 50 halogenated indole derivatives as previously reported [47]. The 3D structures of these compounds were drawn in ChemDraw Ultra version 12.0.2 (Revvity Signals Software, Waltham, MA, USA). The SMILES strings of the compounds were converted into 3D SDF files using the NovaPro web application (https://www.novoprolabs.com/tools/smiles2pdb, accessed on 18 June 2025) and macromolecule energy minimization of the SDF files was performed using the LigPrep module with the OPLS3E force field. Common substructure alignment of all energy-minimized derivatives was carried out using base structure of indole as a reference utilizing the Ligand Alignment tool in Maestro 12.5 (Schrödinger Software Solutions, New York, NY, USA). MICs were converted to pMIC values [-log (MIC)], and a 3D atom-based QSAR model with four partial least squares (PLS) factors was constructed using PHASE (MAESTRO-12.0, Schrödinger Software Solutions, USA). PLS factor 4 was used for QSAR visualization and activity predictions.

### 4.9. Cytotoxicity Assays in HepG2 Cells and a Plant Model

Human HepG2 cells were grown in DMEM + 10% FBS (fetal bovine serum), seeded into 96-well plates. After 24 h, treated with di-halogenated hits (0, 10, 20, 50, 100, and 200 µg/mL) for another 24 h. Cell survival was quantified using MTT (3-(4,5-Dimethylthiazol-2-yl)-2,5-Diphenyltetrazolium Bromide), and absorbance at 570 nm was used to calculate viability relative to untreated controls [58,59].

The phytotoxic effects of di-halogenated hits on seed germination and early growth were assessed using radish seeds (*Raphanus sativus*) [60]. Following five rinses using sterile distilled water, the seeds were thoroughly air-dried. Ten seeds were placed on soft agar Murashige and Skoog (MS) plates (0.7% agar, 0.86 g/L MS) containing di-halogenated hits (0, 5, 10, 20, and 50 µg/mL). Untreated plates served as controls. All plates were incubated at 24 °C for five days, with seed germination rates recorded daily. Each condition was tested in four independent biological replicates, with three technical replicates per experiment.

### 4.10. Statistics

All experiments were conducted using a minimum of two biologically independent cultures, each performed in triplicate. Data are shown as mean values with standard deviations (SD). Group differences were tested by Student’s *t*-test; significance set at *p* ≤ 0.05. Figures were prepared in SigmaPlot version 14.0 (Systat Software, San Jose, CA, USA).

## 5. Conclusions

This work demonstrates that multi-halogenated indoles represent a new and effective chemical class of antifungal and antivirulence agents against drug-resistant *Candida* species. The di-halogenated indoles such as 4,6-dibromoindole and 5-bromo-4-chloroindole exhibited potent fungicidal and antibiofilm activity at low concentrations, suppressed hyphal morphogenesis and induced ROS-mediated oxidative stress. QSAR analysis revealed that halogen substitutions at the C4–C6 positions of indole ring enhances antifungal activity. Both compounds displayed low cytotoxicity toward HepG2 cells and negligible phytotoxicity, indicating a favorable safety margin. These findings reveal that halogenation precisely modulates the electronic and hydrophobic properties of indoles, enhancing their antifungal potency. Multi-halogenation thus represents a rational design approach for developing dual antifungal and antivirulence agents. The low toxicity and simple structure of these indoles suggest strong potential for topical, agricultural, and antifouling applications.

Future work should focus on defining the precise cellular targets of multi-halogenated indoles and optimizing their physicochemical properties to further enhance potency and reduce toxicity. The observed additive interactions with azoles warrant evaluation in combination therapies for azole-resistant infections, including in vivo biofilm and mucosal infection models. In parallel, rational functionalization and QSAR-guided design may yield next-generation indole derivatives with improved pharmacokinetics or broader species coverage.

## Figures and Tables

**Figure 1 ijms-26-10836-f001:**
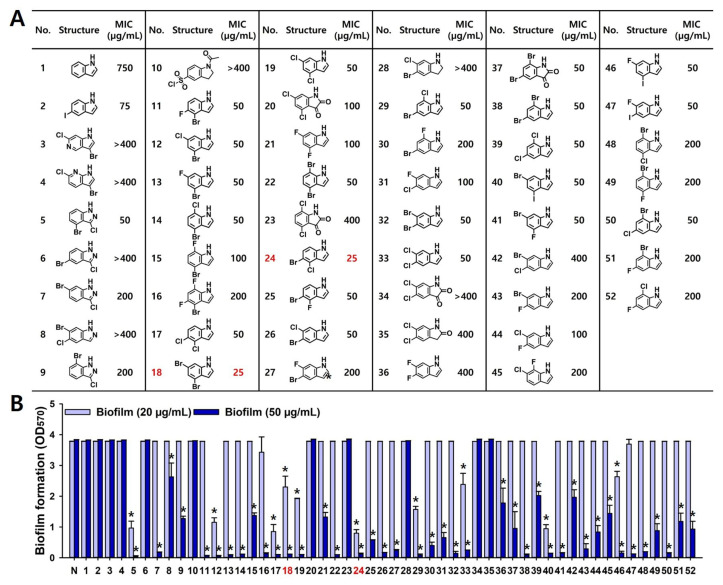
Antifungal and antibiofilm activities of halogenated indoles against *C. albicans.* MICs of 50 halogenated indoles were investigated against *C. albicans* (**A**). Biofilm inhibitory potential was screened at 20 and 50 µg/mL (**B**). #1 (indole) and #2 (5-iodoindole) were used for comparison. Compounds #18 (4,6-dibromoindole) and #24 (5-bromo-4-chloroindole) showed the most potent antifungal and antibiofilm activities. *, *p* < 0.05 compared to untreated.

**Figure 2 ijms-26-10836-f002:**
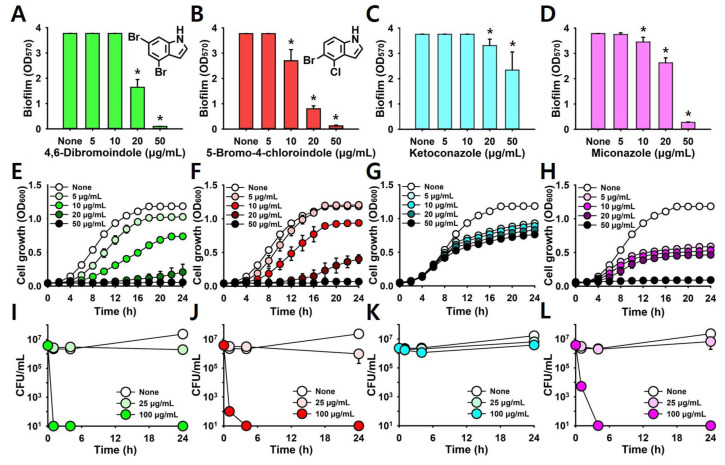
Antibiofilm and fungicidal activities of two active halogenated indoles against *C. albicans.* Dose-dependent inhibition of biofilm formation, planktonic cell growth, and killing kinetics in *C. albicans* with 4,6-dibromoindole (**A**,**E**,**I**) and 5-bromo-4-chloroindole (**B**,**F**,**J**), ketoconazole (**C**,**G**,**K**), and miconazole (**D**,**H**,**L**). Ketoconazole and miconazole were used as controls for comparison. *, *p* < 0.05 compared to untreated.

**Figure 3 ijms-26-10836-f003:**
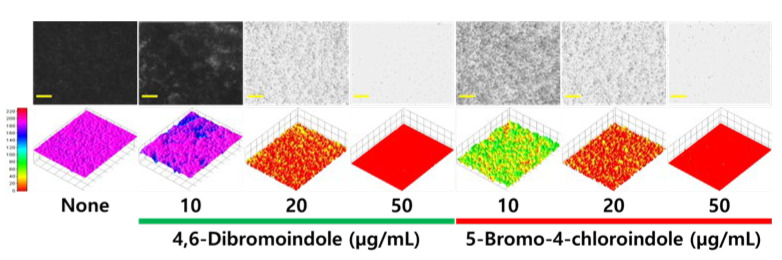
Microscopic analysis of *C. albicans* biofilm. 2D and 3D images showing the biofilm structure of *C. albicans* treated with halogenated indoles. Yellow scale bar represents 200 μm.

**Figure 4 ijms-26-10836-f004:**
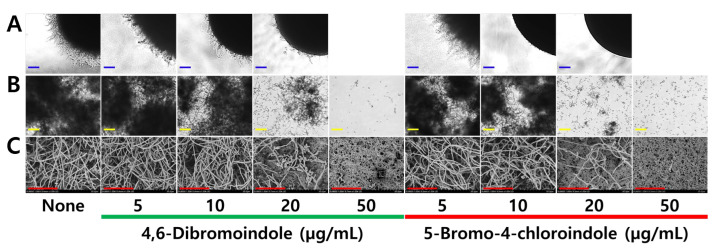
Effects of the halogenated indoles on cell morphology. Colony shape of *C. albicans* (**A**), cell aggregation (**B**) and SEM images of anti-hyphal activities (**C**). Blue, yellow, and red scale bars represent 200 μm, 100 μm and 50 μm, respectively.

**Figure 5 ijms-26-10836-f005:**
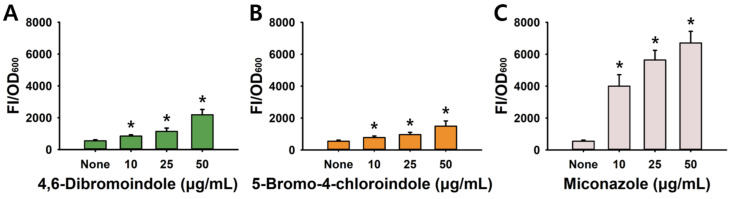
Induction of ROS by halogenated indoles and miconazole in *C. albicans*. Dose-dependent ROS production in *C. albicans* was measured after treatment with 4,6-dibromoindole (**A**), 5-bromo-4-chloroindole (**B**), and miconazole (**C**). ROS levels were quantified as fluorescence intensity normalized to cell density (FI/OD_600_). *, *p* < 0.05 compared to untreated.

**Figure 6 ijms-26-10836-f006:**
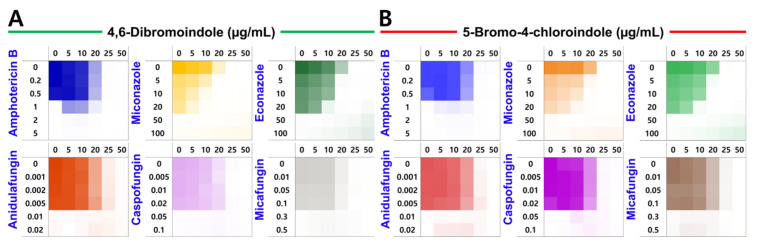
Combinatorial effects with conventional antifungal. MIC heatmaps showing planktonic growth inhibition of *C. albicans* by 4,6-dibromoindole (**A**), 5-bromo-4-chloroindole (**B**) and six conventional antifungals-amphotericin B, miconazole, econazole, anidulafungin, caspofungin, and micafungin.

**Figure 7 ijms-26-10836-f007:**
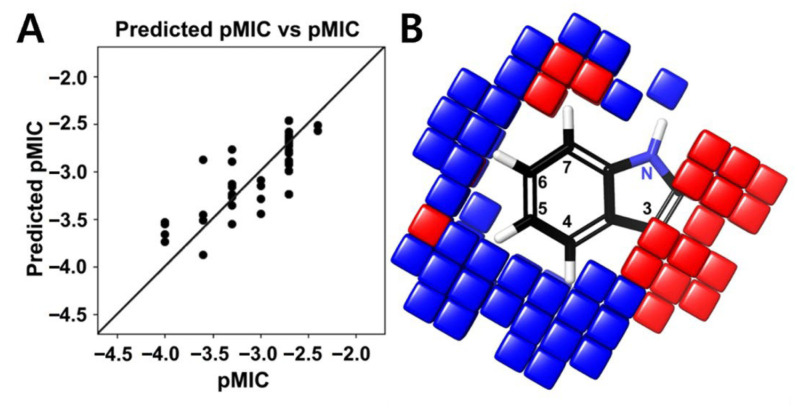
3D-QSAR modeling and predictive pharmacophore analysis. 3D-QSAR model constructed from 50 indole derivatives (**A**). Correlation between observed and predicted pMIC values indicates model reliability. Contour map highlighting favorable (blue) and unfavorable (red) substitution regions (**B**).

**Figure 8 ijms-26-10836-f008:**
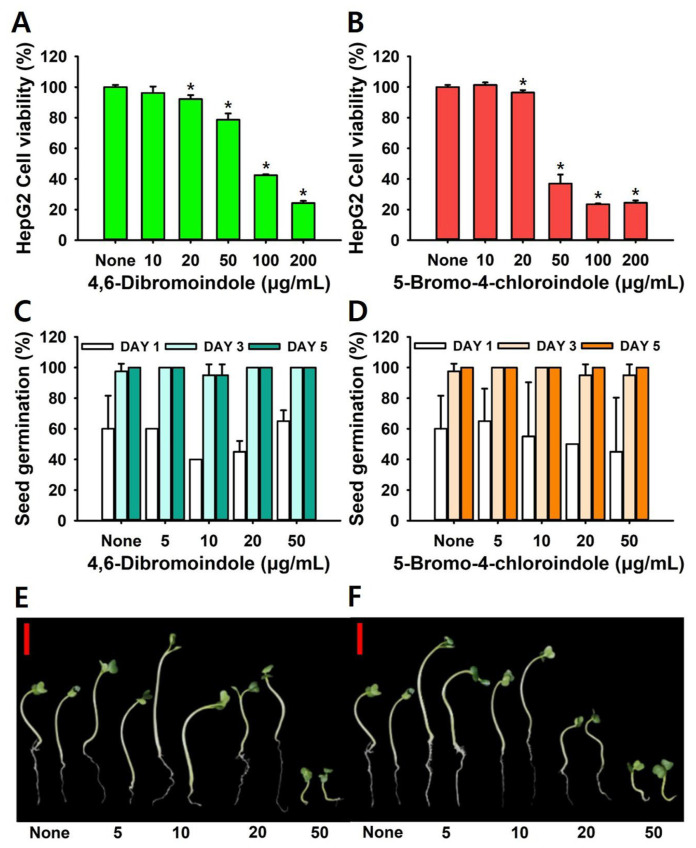
In vivo toxicity assays. HepG2 cell toxicity induced by 4,6-dibromoindole (**A**), 5-bromo-4-chloroindole (**B**). Plant toxicity assay using radish seed germination (**C**,**D**) and plant growth (**E,F**). Red scale bar represents 1 cm. *, *p* < 0.05 compared to untreated.

**Table 1 ijms-26-10836-t001:** MIC of two active compounds against ten *Candida* strains. Ketoconazole and miconazole were used as reference antifungal agents.

MIC (μg/mL)	4,6-Dibromoindole	5-Bromo-4-Chloroindole	Ketoconazole	Miconazole
*C. albicans* DAY 185	25	25	>400	50
*C. albicans* ATCC 10231	25	25	>400	25
*C. auris* KCTC 17809	25	25	50	10
*C. auris* KCTC 17810	10	10	50	10
*C. glabrata* ATCC 2001	25	25	>400	25
*C. glabrata* KCCM 12552	25	50	>400	25
*C. glabrata* KCCM 50701	25	25	>400	25
*C. parapsilosis* ATCC 7330	25	25	>400	25
*C. parapsilosis* ATCC 22019	10	10	25	10
*C. parapsilosis* KCCM 50030	25	25	>400	25

## Data Availability

The original contributions presented in this study are included in the article/Appendix A. Further inquiries can be directed to the corresponding author.

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
