# Peer review of "Antifungal Activities of Multi-Halogenated Indoles Against Drug-Resistant *Candida* Species"

_ijms, 2025, doi:10.3390/ijms262210836_

Round 1

Reviewer 1 Report

Comments and Suggestions for Authors

The objective of the paper is well-defined and demonstrates a detailed and comprehensive study on multi-halogenated indoles against drug-resistant Candida species. The MIC values against the tested pathogens are notably low, indicating strong potential for clinical application. The author has effectively integrated multiple approaches, including structure–activity relationship analysis and biocompatibility evaluation using seed germination assays.

Below are my minor suggestions for improvement:

  1. Lines 281–284: Please include relevant references to support the statements.

  2. Conclusion: Add a few points highlighting the future perspectives of this work.

  3. Figure 5: Include p-values in the figure legend.

  4. Figure 5: Consider separating the ROS results and the combination trial effects into distinct figures for better clarity.

Author Response

The objective of the paper is well-defined and demonstrates a detailed and comprehensive study on multi-halogenated indoles against drug-resistant Candida species. The MIC values against the tested pathogens are notably low, indicating strong potential for clinical application. The author has effectively integrated multiple approaches, including structure–activity relationship analysis and biocompatibility evaluation using seed germination assays.

We would like to thank the reviewer for thorough reading of this manuscript and insightful comments which helped us improve the manuscript's quality and scientific value. We hope our revisions have improved the manuscript to a level of the reviewer’s satisfaction.

Below are my minor suggestions for improvement:

  1. Lines 281–284: Please include relevant references to support the statements.

Thank you for your comment. As suggested, three below relevant references have been included for the statement of the attenuated mammalian toxicity of di-halogenated indole (Line 296).

  1. Wilcken, R.; Zimmermann, M. O.; Lange, A.; Joerger, A. C.; Boeckler, F. M., Principles and applications of halogen bonding in medicinal chemistry and chemical biology. J. Med. Chem. 2013, 56, (4), 1363-1388.
  2. DeGroot, D. E.; Franks, D. G.; Higa, T.; Tanaka, J.; Hahn, M. E.; Denison, M. S., Naturally occurring marine brominated indoles are aryl hydrocarbon receptor ligands/agonists. Chem. Res. Toxicol. 2015, 28, (6), 1176-1185.
  3. King, J.; Woolner, V. H.; Keyzers, R. A.; Rosengren, R. J., Characterization of marine-derived halogenated indoles as ligands of the aryl hydrocarbon receptor. Toxicol. Rep. 2022, 9, 1198-1203.

  1. Conclusion: Add a few points highlighting the future perspectives of this work.

As suggested, the future perspectives of this work has been added as below.

Future work should focus on defining the precise cellular targets of multi-halogenated indoles and optimizing their physicochemical properties to further enhance potency and reduce toxicity. The observed additive interactions with azoles warrant evaluation in combination therapies for azole-resistant infections, including in vivo biofilm and mucosal infection models. In parallel, rational functionalization and QSAR-guided design may yield next-generation derivatives with improved pharmacokinetics or broader species coverage. Given their favorable safety in mammalian and plant models, these compounds also hold potential for medical device coatings and agricultural antifungal applications. (Line 464)

  1. Figure 5: Include p-values in the figure legend.

As suggested, p-values in the Figure 5 has been added.

  1. Figure 5: Consider separating the ROS results and the combination trial effects into distinct figures for better clarity.

Thank you for your comment. As suggested, Figure 5 has been separated into Figure 5 and Figure 6.

Reviewer 2 Report

Comments and Suggestions for Authors

The manuscript by Jeong and co-workers described the Antifungal activities of multi-halogenated indoles against 2 drug-resistant Candida species. Systematically explained the reaction protocol and reported the different characterisation techniques and evaluated for antifungal and antibiofilm activities. Therefore, this protocol is useful for the synthesis of biologically significant molecules; hence, after minor modifications, this could be interesting for readers and may be published in the journal.

Specific comments to the authors

  • Modified the abstract and conclusion based on research findings rather than general statements. Please ensure that your conclusions section emphasises the scientific value added by your paper and the applicability of your findings.
  • In the introduction section, more discussions on currently existing biological applications of indole derivatives should be added and compared with this work. I recommend consulting a few recent articles.
  • What is the source of these indole compounds, and should provide the preparation method.
  • Outline a comprehensive section to discuss the structure-activity relationship and favoured key structural features (pharmacophores) and how far they are influencing the activities under investigation based on experimental and computational studies.
  • The author should discuss and add a figure containing similar inhibitors and their IC50.

Author Response

The manuscript by Jeong and co-workers described the Antifungal activities of multi-halogenated indoles against 2 drug-resistant Candida species. Systematically explained the reaction protocol and reported the different characterisation techniques and evaluated for antifungal and antibiofilm activities. Therefore, this protocol is useful for the synthesis of biologically significant molecules; hence, after minor modifications, this could be interesting for readers and may be published in the journal.

We would like to thank the reviewer for thorough reading of this manuscript and insightful comments which helped us improve the manuscript's quality and scientific value. We hope our revisions have improved the manuscript to a level of the reviewer’s satisfaction.

Specific comments to the authors

  1. Modified the abstract and conclusion based on research findings rather than general statements. Please ensure that your conclusions section emphasises the scientific value added by your paper and the applicability of your findings.

Thank you for your valuable comment. The abstract and conclusion sections have been modified based on research findings and emphasized the scientific value and the applicability. (Line 11-24, 452-471)

  1. In the introduction section, more discussions on currently existing biological applications of indole derivatives should be added and compared with this work. I recommend consulting a few recent articles.

Thank you for your comment. As suggested, Recent studies of biological applications of indole derivatives has been added in the introduction section as follows.

Recent studies (shown below) have highlighted indole as a versatile scaffold in modern drug discovery owing to its broad pharmacological profile, including anticancer, anti-inflammatory, and anti-infective properties (Line 53)

  1. Mo, X.; Rao, D. P.; Kaur, K.; Hassan, R.; Abdel-Samea, A. S.; Farhan, S. M.; Brase, S.; Hashem, H., Indole derivatives: a versatile scaffold in modern drug discovery-an updated review on their multifaceted therapeutic applications (2020-2024). Molecules 2024, 29, (19), 4770.
  2. Mathada, B. S.; Somappa, S. B., An insight into the recent developments in anti-infective potential of indole and associated hybrids. J. Mol. Struct. 2022, 1261, 132808.
  3. Dhiman, A.; Sharma, R.; Singh, R. K., Target-based anticancer indole derivatives and insight into structure‒activity relationship: A mechanistic review update (2018-2021). Acta Pharm. Sin. B 2022, 12, (7), 3006-3027.
  4. Babalola, B. A.; Malik, M.; Olowokere, O.; Adebesin, A.; Sharma, L., Indoles in drug design and medicinal chemistry. Eur J Med Chem Rep 2025, 13, 100252.
  5. Iyer, K. R.; Li, S. C.; Revie, N. M.; Lou, J. W.; Duncan, D.; Fallah, S.; Sanchez, H.; Skulska, I.; Usaj, M. M.; Safizadeh, H.; Larsen, B.; Wong, C.; Aman, A.; Kiyota, T.; Yoshimura, M.; Kimura, H.; Hirano, H.; Yoshida, M.; Osada, H.; Gingras, A. C.; Andes, D. R.; Shapiro, R. S.; Robbins, N.; Mazhab-Jafari, M. T.; Whitesell, L.; Yashiroda, Y.; Boone, C.; Cowen, L. E., Identification of triazenyl indoles as inhibitors of fungal fatty acid biosynthesis with broad-spectrum activity.

  1. What is the source of these indole compounds, and should provide the preparation method.

As mentioned in the Material section (Line 318), all indole compounds (≥ 95 % purity).were purchased from Combi-Blocks (San Diego, California, USA). Their CAS numbers are provided in Supplementary Table 1.

  1. Outline a comprehensive section to discuss the structure-activity relationship and favoured key structural features (pharmacophores) and how far they are influencing the activities under investigation based on experimental and computational studies.

Thank you for your insightful comment. The discussion of SAR has been added as below.

Antifungal screening (Figure 1) and QSAR analyses (Figure 7) revealed that di-halogenation specifically within the C4–C6 region of the indole ring establishes an electronically optimized pharmacophore that enhances membrane-penetrating lipophilicity and redox reactivity, thereby driving potent fungicidal activity and suppression of biofilm-associated morphogenesis. (Line 249-254)

  1. The author should discuss and add a figure containing similar inhibitors and their IC50.

We appreciate this valuable suggestion. In this study, HepG2 cytotoxicity was evaluated only for the two most active di-halogenated indoles. Generating IC₅₀ values for all 50 derivatives would require extensive cell-based experimentation and was not part of the original study. However, we fully agree on the importance of a comprehensive toxicity evaluation, and we have now added a statement in the Discussion that future work will include systematic toxicity profiling of the entire library using both mammalian and plant. Future studies will expand cytotoxicity profiling to the full library of halogenated indoles in mammalian and plant models to better define structure–toxicity relationships and guide scaffold optimization. (Line 301-303)

Reviewer 3 Report

Comments and Suggestions for Authors

The manuscript presents the evaluation of the activity of fifty multi-halogenated indole derivatives as potential antifungal agents. The article addresses the urgent global issue of multidrug-resistant Candida spp., highlighting its relevance. The activity evaluation was performed on ten Candida strains, to which are added several mechanistic investigations and QSAR studies. This article presents an interesting research in the field of medicinal chemistry and should be of great interest to the readers. In my view this paper is ready for publication but may need some minor changes:

  • Line 73 – it says “forty five compounds”, but there are a total of 52 compounds in the table, of which 50 are halogenated indole derivatives.
  • Supplementary figure 1 should be cited in text

Author Response

The manuscript presents the evaluation of the activity of fifty multi-halogenated indole derivatives as potential antifungal agents. The article addresses the urgent global issue of multidrug-resistant Candida spp., highlighting its relevance. The activity evaluation was performed on ten Candida strains, to which are added several mechanistic investigations and QSAR studies. This article presents an interesting research in the field of medicinal chemistry and should be of great interest to the readers. In my view this paper is ready for publication but may need some minor changes:

We would like to thank the reviewer for thorough reading of this manuscript and insightful comments which helped us improve the manuscript's quality and scientific value. We hope our revisions have improved the manuscript to a level of the reviewer’s satisfaction.

Line 73 – it says “forty five compounds”, but there are a total of 52 compounds in the table, of which 50 are halogenated indole derivatives.

Thank you for this comment. Forty five was our mistake. 50 halogenated indole derivatives are correct, which has been corrected. (Line 76)

Supplementary figure 1 should be cited in text

As suggested, Supplementary Figure 1 has been cited. (Lines 167-168)